# Incidence and predictors of cardiovascular disease mortality and all-cause mortality in patients with type II diabetes with peripheral arterial disease

Amaraporn Rerkasem[1,2], Ampica Mangklabruks[3], Supawan Buranapin[3], Kiran Sony[4], Nimit Inpankaew[5], Rath Rerkasem[6], Sasinat Pongtam[1], Kochaphan Phirom[1], Kittipan Rerkasem[1,7,8]*

1 Environmental - Occupational Health Sciences and Non Communicable Diseases Research Center, Research Institute for Health Sciences, Chiang Mai University, Chiang Mai, Thailand, 2 Research Center for Infectious Diseases and Substance Use, Research Institute for Health Sciences, Chiang Mai University, Chiang Mai, Thailand, 3 Department of Internal Medicine, Faculty of Medicine, Chiang Mai University, Chiang Mai, Thailand, 4 Department of Internal Medicine, Chiang Rai Prachanukroh Hospital, Chiang Rai, Thailand, 5 Department of Internal Medicine, Lamphun Hospital, Lamphun, Thailand, 6 Faculty of Medicine, Chiang Mai University, Chiang Mai, Thailand, 7 Clinical Surgical Research Center, Chiang Mai University, Chiang Mai, Thailand, 8 Department of Surgery, Faculty of Medicine, Chiang Mai University, Chiang Mai, Thailand

* rerkase@gmail.com

## Abstract

### Objective

This cohort study estimated the incidence and predictors of cardiovascular disease (CVD) and all-cause mortality among patients with type 2 diabetes mellitus (T2DM) and various stages of peripheral arterial disease (PAD) at the largest tertiary referral hospitals in upper-northern Thailand.

### Methods

This study recruited 278 T2DM and PAD patients for a 7-year cohort study. These patients completed health questionnaires and underwent physical examinations including ankle-brachial index measurements and clinical assessment to determine PAD severity. Mortality endpoints were determined using hospital death registers and national death records. The Cox proportional hazards and subdistribution hazard models were used to estimate PAD's effect on mortality, quantifying the association with hazard ratios (HR) and subdistribution hazard ratios (SHR).

### Results

PAD patients were categorized into three subgroups. Over seven years, the cumulative all-cause mortality rate was 36%, or 6.4 deaths per 100 person-years. Multivariable analysis revealed critical limb ischemia (CLI) patients had significantly higher

**Data availability statement:** All relevant data are within the manuscript and its Supporting Information files.

**Funding:** This study was partially supported by Chiang Mai University (RG14/2567). A.R. was supported by Chiang Mai University (JRCMU2564_078, https://ora.oou.cmu.ac.th/home/). The funders had no role in study design, data collection and analysis, decision to publish, or preparation of the manuscript.

**Competing interests:** The authors have declared that no competing interests exist.

risks of all-cause (HR 5.26, 95%CI 3.10–8.94) and CVD mortality (SHR 6.20, 95%CI 3.20–12.03) compared to their asymptomatic peers. No statistically significant differences in non-CVD mortality were noted across PAD subgroups.

## Conclusion

CLI, chronic kidney disease, and underweight (body mass index < 18.5 kg/m$^2$) emerged as independent mortality predictors. Conversely, asymptomatic PAD patients had a similar overall mortality risk as those with intermittent claudication. These findings highlight the need for risk stratification and patient empowerment to optimize management of these complex conditions.

## Introduction

The global health burden is exacerbated by the high incidence of type 2 diabetes mellitus (T2DM), with Thailand ranking fourth in prevalence in the western Pacific [1]. As of 2021, Thailand saw a significant increase in T2DM cases, rising to 6.1 million adults from 4 million in 2011 [2]. Peripheral arterial disease (PAD), a common complication of T2DM, leads to intermittent claudication (IC), ischemic rest pain, and non-healing foot ulcers, which can result in severe outcomes such as leg infections and amputation [3,4]. Caused primarily by atherosclerosis, PAD affects the medium to large-sized arteries, especially in the lower limbs [5,6]. T2DM increases the risk of cardiovascular disease (CVD), including PAD, which further elevates the overall CVD-related mortality [7]. Critical limb ischemia (CLI), an advanced stage of PAD characterized by rest pain, ischemic ulcer, or gangrene [8], often results in amputation and death [8]. Despite the known prevalence of PAD in diabetics—estimated between 20–30% and associated with a two- to four-fold increased risk of CVD-related mortality [9] —there is a notable gap in detailed research on mortality rates and predictors in Southeast Asia [2,3]. Traditional CVD risk factors like older age, gender, smoking, hypertension, dyslipidemia, and obesity, are established predictors of mortality in PAD patients [10]. Yet, the specific impact of these factors, combined with T2DM and PAD, on mortality remains underexplored. This study aims to address this gap by examining CVD-related and all-cause mortality rates and identifying prognostic predictors among patients with T2DM and PAD.

## Materials and methods

### Study design

The study was conducted from May 1, 2014, to December 31, 2021, and included participants aged over 45 years with T2DM from the three largest tertiary referral hospitals in Northern Thailand: Maharaj Nakorn Chiang Mai Hospital, Chiangrai Prachanukroh Hospital, and Lamphun Hospital. Recruitment began on May 1, 2014, and was completed by February 26, 2015. The study aimed to evaluate the long-term impact of PAD on mortality in this population. Written informed consent was obtained from all participants prior to enrollment.

## Study population

From the initial screening of 2,247 T2DM patients for PAD at outpatient clinics, 278 individuals who met the inclusion criteria were recruited for this long-term cohort study. Inclusion criteria required were a diagnosis of T2DM, age ≥ 45, and PAD confirmed by symptoms or an ankle-brachial index (ABI) ≤0.9. Imaging techniques, such as duplex ultrasound, digital subtraction angiography (DSA), computed tomography angiography (CTA), or magnetic resonance angiography (MRA) were used for confirmation in selected cases when clinically indicated. All participants were required to have been under hyperglycemic control for at least six months before enrollment.

The exclusion criteria were as follows: patients with a new diagnosis of cerebrovascular or coronary events within three months of enrollment to ensure baseline measurements reflected stable disease states rather than acute post-event changes. Individuals with stage III-IV malignancy or terminal cancer and a life expectancy less than one year, as well as those with active HIV infection or AIDS-related vasculopathy were also excluded. Furthermore, patients requiring scheduled vascular surgery within the following six months—such as those with symptomatic abdominal aortic aneurysms or symptomatic thoracic aortic aneurysms—were not included. Finally, those with systemic infections affecting the vasculature, such as fungal arteritis, were excluded from study.

Due to the inclusion criteria and clinical characteristics of the study population, the anatomical location of PAD was not uniformly determined. Advanced imaging techniques (e.g., DSA, CTA, MRA) were reserved for cases where revascularization was clinically necessary, in line with European guidelines [11]. Initial assessments relied on physical examination, ABI measurements, and duplex ultrasonography, which are sufficient for diagnosing PAD but not for detailed anatomical localization of lesions.

The study was approved by the human experimentation ethics committee of the Faculty of Medicine, Chiang Mai University (ethics reference number 2564–08149), adhering to the Declaration of Helsinki, with all participants providing written informed consent.

## Data collection

At enrollment, comprehensive data were collected from each participant, including baseline demographic details, vital signs, physical examination findings, ABI test results, and CVD history. Recent data pertinent to CVD prognosis—gathered from medical records within the last three months—covered lipid profiles, fasting blood glucose levels, hemoglobin A1c (HbA1c), estimated Glomerular Filtration Rate (eGFR) from biochemical tests, medication details, and radiographic examination findings. Trained nurses were responsible for participant monitoring, data completion, and record submission to Chiang Mai University's Research Institute for Health Sciences.

## Primary endpoint

The primary endpoints were causes of death, ascertained from hospital records and national death registration systems up to the end of 2021, which was the end of the follow-up period. For participants lost to follow-up, primary cause of death information was sourced from the Ministry of Interior's death registration system. Causes of death were categorized into CVD or non-CVD causes, employing the ICD-10-TM 2016 diagnosis codes to specifically categorize CVD mortality into acute myocardial infarction (MI), sudden cardiac arrest, heart failure (I20-I50), stroke (I60-I69), and PAD (E11.5, I70.2, I79.2, I99, L97).

## Statistical analysis

Patients were stratified into three groups based on PAD severity at enrollment: CLI, IC, and asymptomatic, examining CVD mortality, non-CVD mortality, and survivors. Descriptive statistics for categorical variables were presented as proportions and percentages, while continuous variables reported as mean and standard deviation for normally

distributed data and median and interquartile range (IQR) for skewed data. Group comparisons were performed using one-way analysis of variance for numerical variables with normal distribution, and the Pearson chi-squared test, or Fisher's exact test categorical variables, particularly when sample sizes were small (n ≤ 5). Survival analysis techniques were employed to estimate time from PAD diagnosis to death or end of follow-up, using life tables and Kaplan-Meier estimates. Cox proportional hazards models, with the assumption of constant hazard ratios over time verified by Schoenfeld residual tests, were used to evaluate the association between variables and all-cause mortality. Recognizing non-CVD mortality as competing events, we conducted a competing risk survival analysis, as described by Austin et al. [12], using Fine-Gray subdistribution hazard models to specifically assess factors influencing CVD mortality.

A backward stepwise selection procedure was utilized in multivariable survival analysis to identify significant predictors of mortality, removing variables with a p-value ≥ .1 at each step and retaining those with a p-value < .05. This included maintaining the entirety of a categorical variable if any of its categories were significant (p < .05), to accurately reflect its contribution to mortality risk. Additionally, we examined interaction effects and assessed multicollinearity among predictors using variance inflation factors to ensure the robustness of our model. Statistical analyses were conducted using Stata version 17 (StataCorp, Texas, USA), with a significance level set at p-value < .05.

## Results

This study followed 278 patients with T2DM and PAD over a seven-year period, with a median follow-up of 81 months (IQR 57 − 82 months). During this period, 100 participants (36%) died. The baseline characteristics of the cohort, divided into survivors, CVD mortality, and non-CVD mortality groups, are presented in Table 1. The mean age at baseline for survivors was 64.9 ± 10.1 years, while those in the CVD mortality and non-CVD mortality groups were older, with mean ages of 69.6 ± 9.2 years and 69.1 ± 9.4 years, respectively. The proportion of males was higher in the non-CVD mortality group (57.1%) compared to the CVD mortality (39.7%) and survivor groups (31.5%).

Table 1 also shows that the prevalence of asymptomatic PAD was highest among survivors (93.8%). The prevalence of chronic ulcers and lower limb gangrene was higher in the group of CVD mortality (29.3%) compared to non-CVD mortality (9.5%) and survivors (2.2%). Tobacco smoking, chronic kidney disease (CKD) stage ≥3, and insulin therapy were more common among the deceased than survivors. Of the patients with CKD, 6 patients were on maintenance hemodialysis at baseline. Among these hemodialysis patients, 6 died during follow-up (3 from CVD-related causes, 3 from non-CVD causes). Survivors had a higher mean BMI and waist circumference and were more likely to be on insulin-sensitizing therapy. There were no significant differences among the three groups in terms of achieving adequate blood pressure control (<130/80 mmHg), optimal low-density lipoprotein cholesterol levels (<70 mg/dL), and glycemic control targets (HbA1c<7% or fasting blood sugar <130 mg/dL), as detailed in S1 Table. The history of MI at baseline was more common in the CVD mortality group compared to non-CVD mortality and survivor groups.

Table 2 lists the causes of death, identifying CVD as the leading cause (58% of deaths), with sudden cardiac arrest as the most common etiology among CVD mortality. End-stage renal disease was a frequent cause among non-CVD mortality. Fig 1 illustrates the distribution of CVD and non-CVD mortality over the follow-up period, indicating a significant difference in the proportion of CVD to non-CVD mortality (exact p = .041). The overall mortality rate was 6.4 deaths per 100 person-years (95% confidence interval (CI) 5.3, 7.8). Table 3 stratifies the mortality rate based on three subgroups of patients with PAD: (i) 239 asymptomatic patients; (ii)12 patients with IC; and (iii) 27 patients with CLI. The CLI subgroup demonstrated a higher 7-year mortality rate compared to their counterparts with IC (14.6 deaths per 100 person-years, 95% CI 1.0, 28.3) and asymptomatic PAD (20.4 deaths per 100 person-years, 95% CI 9.7, 31.1). Fig 2 depicts all-cause mortality trends among the 278 patients, categorized into the same three subgroups, across the follow-up period. The 7-year mortality rates varied: 30.1% in asymptomatic patients (comprising 38 CVD mortality, 34 non-CVD mortality),

**Table 1. Baseline characteristics of participants with T2DM and PAD and outcomes after 7 years follow-up (N = 278).**

| Variables | T2DM patients | | | p-value |
|---|---|---|---|---|
| | Survivors | Non-CVD mortality | CVD mortality | |
| n | 178 | 42 | 58 | |
| **Demographics** | | | | |
| Age - years | 64.9 ± 10.1 | 69.1 ± 9.4 | 69.6 ± 9.2 | **.002** |
| Male ratio | 56 (31.5%) | 24 (57.1%) | 23 (39.7%) | **.007** |
| **Educational attainment** | | | | .25 |
| Lower than high school | 111 (62.4%) | 31 (73.8%) | 41 (70.7%) | |
| High school or higher | 67 (37.6%) | 11 (26.2%) | 17 (29.3%) | |
| **Cardiovascular risk factors** | | | | |
| Smoker | | | | **.025** |
| Never | 127 (71.4%) | 22 (52.4%) | 35 (60.3%) | |
| Former | 36 (20.2%) | 16 (38.1%) | 21 (36.2%) | |
| Current | 15 (8.4%) | 4 (9.5%) | 2 (3.4%) | |
| Body mass index - kg/m$^2$ | 26.6 ± 6.2 | 24.4 ± 4.8 | 22.6 ± 5.0 | **<.001** |
| Waist circumference - cm | | | | |
| Male | 93.7 ± 13.4 | 87.2 ± 13.0 | 88.2 ± 9.0 | .066 |
| Female | 90.5 ± 13.0 | 87.3 ± 11.4 | 80.7 ± 11.5 | **.001** |
| Hypertension | 152 (85.4%) | 39 (92.9%) | 51 (87.9%) | .49 |
| Dyslipidemia | 158 (88.8%) | 33 (78.6%) | 49 (84.5%) | .20 |
| CKD stage ≥3 [a] | 29 (16.3%) | 23 (54.8%) | 32 (55.2%) | **<.001** |
| 1st degree relatives premature ASCVD | 13 (7.3%) | 3 (7.1%) | 2 (3.4%) | .64 |
| 10-year ASCVD risk score [b] | 23.0 ± 3.7 | 24.1 ± 3.6 | 23.0 ± 4.0 | .20 |
| Peripheral arterial disease | | | | **<.001** |
| Asymptomatic | 167 (93.8%) | 34 (80.9%) | 38 (65.5%) | |
| IC | 6 (3.4%) | 3 (7.1%) | 3 (5.2%) | |
| Rest pain | 1 (0.6%) | 1 (2.4%) | 0 (0.0%) | |
| Chronic ulcer | 4 (2.2%) | 4 (9.5%) | 13 (22.4%) | |
| Gangrene | 0 (0.0%) | 0 (0%) | 4 (6.9%) | |
| **History of cardiovascular events** | | | | |
| MI | 9 (5.1%) | 1 (2.4%) | 8 (13.8%) | **.032** |
| Ischemic stroke | 9 (5.1%) | 1 (2.4%) | 4 (6.9%) | .56 |
| Transient ischemic attack | 0 (0.0%) | 1 (2.4%) | 0 (0.0%) | .15 |
| **Current medications used** | | | | |
| Antithrombotic | 119 (66.8%) | 28 (66.7%) | 43 (74.1%) | .57 |
| Single antiplatelet [c] | 110 (61.8%) | 24 (57.1%) | 38 (65.5%) | .70 |
| Dual antiplatelet [d] | 5 (2.8%) | 3 (7.1%) | 5 (8.6%) | .10 |
| Oral anticoagulants [e] | 5 (2.8%) | 2 (4.8%) | 2 (3.4%) | .70 |
| Beta-blocker | 50 (28.1%) | 11 (26.2%) | 15 (25.9%) | .93 |
| RAAS inhibitors [f] | 118 (66.3%) | 23 (54.8%) | 30 (51.7%) | .087 |
| Calcium channel blockers | 75 (42.1%) | 18 (42.9%) | 28 (48.3%) | .71 |
| Diuretics | 61 (34.4%) | 19 (45.2%) | 23 (39.7%) | .37 |
| Statin | 131 (73.6%) | 27 (64.3%) | 41 (70.7%) | .48 |
| Insulin analog | 59 (33.2%) | 23 (54.8%) | 22 (37.9%) | **.034** |
| Insulin secretagogue [g] | 91 (51.1%) | 14 (33.3%) | 25 (43.1%) | .095 |
| Insulin sensitizing [h] | 112 (62.9%) | 15 (35.7%) | 25 (43.1%) | **.001** |
| DDP-4 inhibitors | 12 (6.7%) | 0 (0.0%) | 4 (6.9%) | .23 |

*(Continued)*

**Table 1.** (Continued)

Data are presented as mean ± standard deviation (SD) or n (%).

[a] CKD is defined as eGFR < 60 ml/min/1.73 m² for ≥ 3 months and/or albuminuria. An isolated eGFR < 60 ml/min/1.73 m² was considered CKD only with additional evidence of kidney damage. [b] Assessment of cardiovascular disease risk by the Rama-EGAT heart score in Thai population. [c,d] Included aspirin, P2Y12 inhibitors, PDE3 inhibitor and combined for dual antiplatelet therapy. [e] Included warfarin therapy. [f] Included ACEI, ARBs, and combined ACEI/ARB therapy. [g] Consisted of sulfonylureas groups. [d] Consisted of biguanides and thiazolidinediones.

Abbreviation: T2DM, type 2 diabetes mellitus; PAD, peripheral arterial disease; CKD, chronic kidney disease; ASCVD, atherosclerotic cardiovascular diseases; IC, intermittent claudication; MI, myocardial infarction; PDE3, phosphodiesterase-3; RAAS, Renin-angiotensin-aldosterone system; ACEI, angiotensin converting enzyme inhibitors; ARBs, angiotensin receptor blockers; DDP-4 inhibitor, Dipeptidyl Peptidase-4 Inhibitor.

**Table 2. Clinical endpoints: Causes of death in participants with T2DM and PAD over a 7-years follow-up period (N = 278).**

| Endpoints | All patients |
| --- | --- |
| CVD mortality | 58 |
| Acute MI | 11 (19.0%) |
| Hemorrhagic stroke | 4 (6.9%) |
| Ischemic stroke | 8 (13.8%) |
| Sudden cardiac arrest | 17 (29.3%) |
| Other cardiovascular mortality | 18 (31.0%) |
| - Heart failure | 10 (17.2%) |
| - Limb ischemia with complication | 8 (13.8%) |
| Non-CVD mortality | 42 |
| End-stage renal disease | 14 (33.3%) |
| Acute renal failure | 2 (4.8%) |
| Malignancy | 4 (9.5%) |
| Gastrointestinal hemorrhage | 2 (4.8%) |
| Cirrhosis | 1 (2.4%) |
| Chronic Obstructive Pulmonary Disease | 3 (7.1%) |
| Pneumonia | 5 (11.9%) |
| Sepsis | 11 (26.2%) |

50.0% in patients with IC (including 3 CVD mortality, 3 non-CVD mortality), and 81.5% in CLI patients (consisting of 17 CVD mortality and 5 non-CVD mortality). Statistically significant differences in the overall probability of death among the three groups were identified (log-rank for overall death, p-value < .001).

Table 4 examined baseline variables associated with all-cause mortality, identifying CLI, being underweight (BMI < 18.5 kg/m²), CKD stage ≥3 and using insulin sensitizer therapy as independent predictors. In Table 5, the proportional subdistribution hazard model shows that CLI, being underweight, and having CKD stage ≥3 were also significantly associated with CVD mortality.

Fig 3 displays cumulative incidence curves for CVD mortality within the PAD subgroups, with Wald test results indicating a higher rate of CVD mortality in CLI patients compared asymptomatic and IC patients (p < .001 and p = .036, respectively). The rate of CVD mortality did not significantly differ between IC and asymptomatic patients. There was no significant difference in non-CVD mortality among PAD subgroups.

## Discussion

This cohort study evaluates the impact of PAD on seven-year mortality rate among T2DM patients, underscoring the critical interplay between PAD severity and T2DM on patient outcomes. Our findings demonstrate a discernibly higher

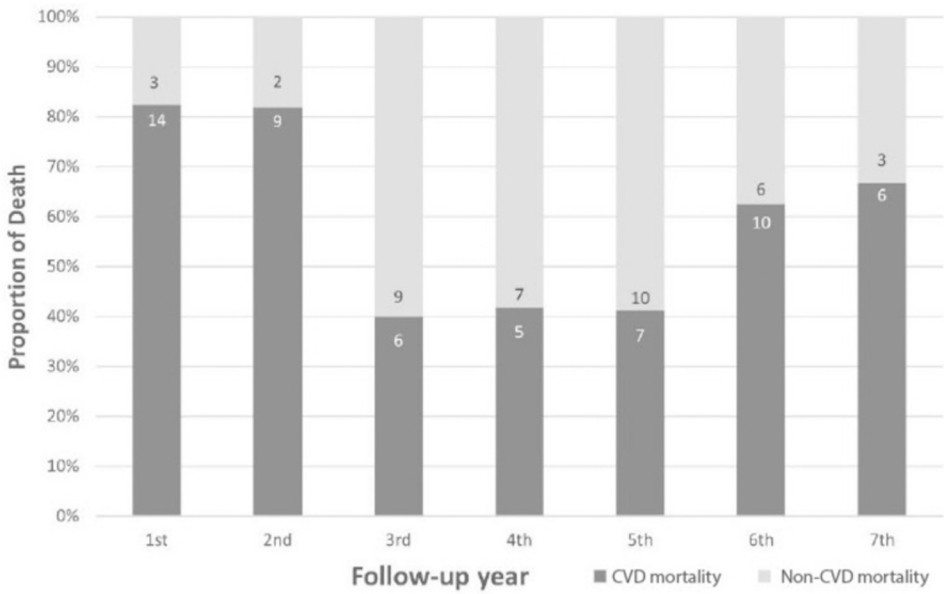

**Fig 1. CVD and non-CVD mortality by year of follow-up.** Numbers in bars indicate the number of deaths.

**Table 3. Crude rate (per 100 person-years) for all-cause mortality, CVD mortality and non-CVD mortality in participants with T2DM and PAD by PAD severity at baseline, 2014-2020.**

| PAD groups at baseline | N | All-cause mortality | CVD mortality | Non-CVD mortality |
|---|---|---|---|---|
| | | Incidence (95%CI) | Incidence (95%CI) | Incidence (95%CI) |
| Asymptomatic | 239 | 5.1 (4.0, 6.4) | 2.7 (2.0, 3.7) | 2.4 (1.7, 3.4) |
| IC | 12 | 10.9 (4.9, 24.3) * | 5.5 (1.8, 17.0) | 5.5 (1.8, 17.0) |
| CLI | 27 | 25.5 (16.8, 38.7) ** † | 19.7 (12.2, 31.7) ** † | 5.8 (2.4, 13.9) |

* p = .050, ** p < .050 compared to asymptomatic PAD. † p < .050 compared to intermittent claudication PAD.

IC, intermittent claudication; CLI, critical limb ischemia refers to ischemic rest pain, gangrene, or ulcer in one or both legs attributable to PAD; CVD, cardiovascular disease; T2DM, type 2 diabetes mellitus; PAD, peripheral arterial disease.

mortality due to CVD in patients with concomitant PAD and T2DM, accentuating the intertwined pathophysiology of these conditions [5]. Notably, the presence of symptomatic PAD at baseline emerged as a significant predictor of elevated CVD mortality. Among the different PAD stages, CLI was the most pronounced predictor of increased mortality, with CLI patients experiencing the highest rates of CVD mortality (81.5%) compared to those presenting with IC (50.0%) and asymptomatic patients (30.1%). These results highlight the prognostic value of PAD severity in T2DM patients.

Our analysis reveals a startling finding that nearly half of the T2DM patients diagnosed with advanced PAD stages die within three years of diagnosis. This mortality risk is consistent with previous studies that highlight the poor prognosis following a CLI diagnosis [13,14]. The high mortality rate underscores the life-threatening nature of PAD when combined with T2DM and emphasizes the urgent need for aggressive clinical intervention in this patient population.

Mortality rates of PAD often vary across different studies, depending on the age of the participants. For instance, in studies involving individuals with an average age younger than 60 report an overall mortality rate ranging from 1.8 to 2.5 per 100 person-years [15,16]. Conversely, research on older individuals with PAD tends to report higher mortality rates,

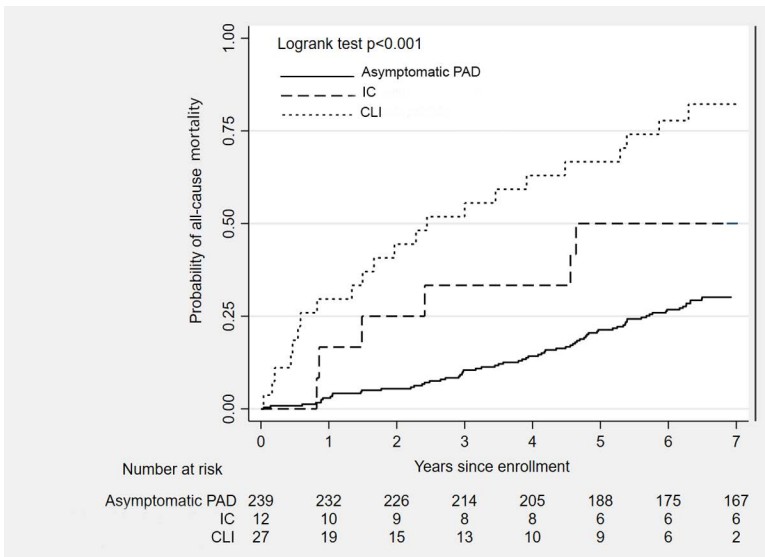

**Fig 2. Kaplan-Meier failure curve shows the cumulative probability for all-cause mortality for participants with T2DM and PAD, stratified by PAD severity at baseline.** Patients with CLI had higher mortality rate than the asymptomatic group (p < .001) but marginally higher for patients with IC (p = .069). A slightly higher rate of dying during follow-up for those patients with IC compared to those with asymptomatic PAD was observed (p = .052).

ranging from 6 to 10 per 100 person-years [15,17]. This disparity may be attributed to a higher tendency to have cardio-vascular events in older age groups. Our study participants were predominantly elderly, with 72% older than 60 years, which may explain the notably high mortality rate observed throughout the entire follow-up period.

Patients with IC typically experience higher mortality rates than those without PAD [18]. However, limited research explores mortality variations across PAD stages. Our study reveals that IC patients exhibit comparable all-cause and CVD-related mortality rates to those with asymptomatic PAD, suggesting that T2DM patients with asymptomatic PAD should not be underestimated. This finding is consistent with previous studies involving hemodialysis patients [19] and elderly individuals in Germany [20], which found no significant difference in mortality risk between IC and asymptomatic PAD when considering all-cause or CVD mortality alone.

Despite IC being the classic PAD symptom, it is noteworthy that 70–80% of diabetic PAD cases are often asymptom-atic due to complications from diabetic peripheral neuropathy, which alters pain sensation [21]. Atypical leg pain often overlaps with classic symptoms of IC, leading to frequent misclassification as asymptomatic [22]. Furthermore, a seden-tary lifestyle, one of the significant risk factors for T2DM [23], and insufficient physical activity typically does not induce claudication symptoms [22]. Consequently, the proportion of diabetic PAD patients experiencing IC complaints is rela-tively low. Even without symptoms like IC, diabetic patients with asymptomatic PAD still have high prevalence of athero-sclerosis in other vascular beds, including coronary artery and cerebrovascular disease, which can increase their risk of cardiovascular-related mortality [24]. Consequently, diabetic patients with asymptomatic PAD in this study exhibit all-cause mortality and CVD mortality rates comparable to those with IC in PAD. Identifying asymptomatic or atypically symptomatic PAD in diabetic patients remains challenging.

CKD stage ≥3 concomitant with T2DM and PAD increased the risk of mortality fourfold compared to normal kidney function. In our study, baseline CKD stage ≥3 emerged as a significant independent predictor of total and CVD-specific mortality among diabetic PAD patients. This finding is consistent with a prior meta-analysis in PAD patients, showing that CKD presence increases mortality risk compared to those without CKD [25]. Regardless of the stage of decreased kidney function, systemic atherosclerotic risk rises [26]. Even mild to moderate kidney disease independently associates with

**Table 4. Cox regression analysis of risk factors associated with all-cause mortality at 7 years follow-up in participants with T2DM and PAD.**

| Baseline characteristics | Univariable analysis HRᵇ (95% CI) | p-value | Multivariable analysis ᵃ HRᵇ (95% CI) | p-value |
|---|---|---|---|---|
| Age (every 5-year increased) | 1.18 (1.08, 1.30) | **.001** | | |
| Male *vs* female | 1.66 (1.12, 2.46) | **.012** | | |
| Former/current smoker *vs* never smoker | 1.64 (1.11, 2.44) | **.014** | | |
| PAD: Asymptomatic | Ref | | Ref | |
| IC | 2.20 (0.96, 5.06) | .064 | 2.08 (0.86, 5.00) | .10 |
| CLI | 5.36 (3.31, 8.68) | **<.001** | 5.26 (3.10, 8.94) | **<.001** |
| Prior MI | 1.72 (0.86, 3.40) | .12 | | |
| Previous stroke or TIA | 1.11 (0.49, 2.54) | .80 | | |
| High 10-year ASCVD risk score ᶜ | 1.21 (0.44, 3.28) | .71 | | |
| Normal weight (BMI 18.5–24.9 kg/m²) | Ref | | Ref | |
| Underweight (BMI < 18.5 kg/m²) | 2.45 (1.36, 4.38) | **.003** | 2.92 (1.62, 5.26) | **<.001** |
| Overweight/obese (BMI ≥ 25 kg/m²) | 0.62 (0.40, 0.96) | **.033** | 1.15 (0.72, 1.84) | .57 |
| CKD stage ≥3 ᵈ | 3.87 (2.61, 5.75) | **<.001** | 4.22 (2.78, 6.42) | **<.001** |
| Hypertension | 1.44 (0.75, 2.76) | .28 | | |
| Dyslipidemia | 0.63 (0.40, 1.27) | .076 | | |
| Poor glycemic control ᵉ | 0.64 (0.43, 0.94) | **.025** | | |
| Single antiplatelet ᶠ | 1.02 (0.68, 1.52) | .94 | | |
| Dual antiplatelet ᵍ | 2.11 (1.02, 4.35) | **.043** | | |
| Oral anticoagulants ʰ | 1.44 (0.53, 3.93) | .47 | | |
| RAAS inhibitors ⁱ | 0.63 (0.42, 0.93) | **.020** | | |
| Statin | 0.78 (0.51, 1.18) | .24 | | |
| Insulin analog | 1.45 (0.98, 2.16) | .063 | | |
| Insulin secretagogue ʲ | 0.65 (0.44, 0.97) | **.036** | | |
| Insulin sensitizer ᵏ | 0.47 (0.31, 0.70) | **<.001** | 0.66 (0.44, 0.99) | **.047** |
| DDP-4 inhibitors | 0.66 (0.24, 1.79) | .41 | | |

ᵃ Variable with p-value < .2 in univariate were included in multivariate analysis; final multivariable model was derived using backward stepwise selection procedure with retention threshold of p < 0.05. ᵇ Cox proportional hazard models for cause-specific hazard ratio (HR). ᶜ Prevalence of significant ASCVD in high-risk subjects (RAMA-EGAT score ≥ 17) was more than 10%. ᵈ Defined as eGFR < 60 ml/min/1.73 m² for ≥ 3 months and/or albuminuria. An isolated eGFR < 60 ml/min/1.73 m² was considered CKD only with additional evidence of kidney damage. ᵉ Poor glycemic control with T2DM was HbA1c ≥ 7% or fasting blood sugar >130 mg/dL. ᶠ,ᵍ Included aspirin, P2Y12 inhibitors, PDE3 inhibitor and combined for dual antiplatelet therapy. ʰ Included warfarin therapy. ⁱ Included ACEI and ARBs or combined ACEI/ARB therapy. ʲ Consisted of sulfonylureas groups. ᵏ Consisted of biguanides and thiazolidinediones.

Abbreviations: T2DM, type 2 diabetes mellitus; PAD, peripheral arterial disease; CI, confidence interval; IC: intermittent claudication; CLI, critical limb ischemia defined by rest pain, nonhealing ulcer or gangrene; MI, myocardial infarction; TIA, transient ischemic attack; ASCVD, atherosclerotic cardiovascular diseases; BMI, body mass index is calculation using a patient's weight (kg)/height(m)2; CKD, chronic kidney disease; PDE3, phosphodiesterase-3; RAAS, Renin-angiotensin-aldosterone system; ACEI, angiotensin converting enzyme inhibitors; ARBs, angiotensin receptor blockers; DDP-4 inhibitor, Dipeptidyl Peptidase-4 Inhibitor.

Significant p-values (<.050) are in bold.

PAD development, and the presence of albuminuria notably links to PAD-related amputation [27]. In our cohort study, we observed a higher prevalence of CKD stage ≥3 in T2DM patients with PAD, especially in older age groups. This elevated prevalence was particularly significant in patients exhibiting symptomatic PAD, most notably in those diagnosed with CLI. Furthermore, the initial prevalence of CKD stage ≥3 was documented at 55% in the deceased group, in contrast to 16% in the survivor group.

This cohort study examined the differential impact of BMI on mortality among patients with T2DM and PAD. Notably, a BMI below 18.5 kg/m² emerged as an independent predictor of mortality, associated with a quadrupled risk of both

**Table 5. Competing risk regression analysis of risk factors associated with CVD mortality at 7 years follow-up in participants with T2DM and PAD.**

| Baseline characteristics | Univariable analysis SHR[b] (95% CI) | p-value | Multivariable analysis [a] SHR[b] (95% CI) | p-value |
|---|---|---|---|---|
| Age (every 5-year increased) | 1.18 (1.05, 1.33) | **.007** | | |
| Male *vs* female | 1.13 (0.67, 1.91) | .65 | | |
| Former/current smoker *vs* never smoker | 1.35 (0.80, 2.29) | .26 | | |
| PAD: Asymptomatic | Ref | | Ref | |
| IC | 1.72 (0.52, 5.60) | .37 | 1.31 (0.27, 6.30) | .73 |
| CLI | 6.57 (3.57, 12.08) | **<.001** | 6.20 (3.20, 12.03) | **<.001** |
| Prior MI | 2.83 (1.32, 6.04) | **.007** | | |
| Previous stroke or TIA | 1.32 (0.49, 3.58) | .58 | | |
| High 10-year ASCVD risk score [c] | 0.90 (0.26, 3.08) | .87 | | |
| Normal weight (BMI 18.5–24.9 kg/m²) | Ref | | Ref | |
| Underweight (BMI < 18.5 kg/m²) | 3.73 (1.92, 7.21) | **<.001** | 4.06 (2.02, 8.18) | **<.001** |
| Overweight/obese (BMI ≥ 25 kg/m²) | 0.56 (0.30, 1.02) | .059 | 1.05 (0.53, 2.06) | .89 |
| CKD stage ≥3 [d] | 3.41 (2.04, 5.71) | **<.001** | 3.78 (2.22, 6.46) | **<.001** |
| Hypertension | 1.10 (0.50, 2.41) | .81 | | |
| Dyslipidemia | 0.84 (0.41, 1.71) | .62 | | |
| Poor glycemic control [e] | 0.59 (0.35, 0.99) | **.045** | | |
| Single antiplatelet [f] | 1.20 (0.70, 2.05) | .51 | | |
| Dual antiplatelet [g] | 2.21 (0.87, 5.58) | .094 | | |
| Oral anticoagulants [h] | 1.22 (0.25, 5.88) | .81 | | |
| RAAS inhibitors [i] | 0.63 (0.38, 1.06) | .080 | | |
| Statin | 0.92 (0.52, 1.63) | .77 | | |
| Insulin analog | 1.01 (0.60, 1.72) | .96 | | |
| Insulin secretagogue [j] | 0.83 (0.50, 1.40) | .49 | | |
| Insulin sensitizer [k] | 0.57 (0.34, 0.96) | **.033** | | |
| DDP-4 inhibitors | 1.28 (0.45, 3.60) | .64 | | |

[a] Variable with p-value < .2 in univariate were included in multivariate analysis; final multivariable model was derived using backward stepwise selection procedure with retention threshold of p < 0.05. [b] Fine-Gray regression for competing risk analysis (non-CVD mortality is a competing risk event for CVD mortality) were presented in subdistribution hazards (SHR). [c] Prevalence of significant ASCVD in high-risk subjects (RAMA-EGAT score ≥ 17) was more than 10%. [d] Defined as eGFR < 60 ml/min/1.73 m² for ≥ 3 months and/or albuminuria. An isolated eGFR < 60 ml/min/1.73 m² was considered CKD only with additional evidence of kidney damage. [e] Poor glycemic control with T2DM was HbA1c ≥ 7% or fasting blood sugar >130 mg/dL. [f, g] Included aspirin, P2Y12 inhibitors, PDE3 inhibitor and combined for dual antiplatelet therapy. [h] Included warfarin therapy. [i] Included ACEI and ARBs or combined ACEI/ARB therapy. [j] Consisted of sulfonylureas groups. [k] Consisted of biguanides and thiazolidinediones.

Abbreviations: CI, confidence interval; IC: intermittent claudication; CLI, critical limb ischemia defined by rest pain, nonhealing ulcer or gangrene; MI, myocardial infarction; TIA, transient ischemic attack; PAD, peripheral arterial disease; ASCVD, atherosclerotic cardiovascular diseases; BMI, body mass index is calculation using a patient's weight (kg)/height(m2)2; CKD, chronic kidney disease; PDE3, phosphodiesterase-3; RAAS, Renin-angiotensin-aldosterone system; ACEI, angiotensin converting enzyme inhibitors; ARBs, angiotensin receptor blockers; DDP-4 inhibitor, Dipeptidyl Peptidase-4 Inhibitor.

Significant p-values (<.050) are in bold.

CVD-related and all-cause mortality compared to patients with a normal BMI range, while the association between overweight or obesity (BMI > 24.9 kg/m²) and increased mortality risk did not achieve statistical significance. These findings partially align with existing meta-analyses, which have reported a diminished all-cause mortality risk among overweight patients with PAD relative to their normal-weight counterparts [28]. This reverse causality phenomenon, where a lower BMI is a consequence of underlying health conditions negatively affecting prognosis rather than solely a predictor of

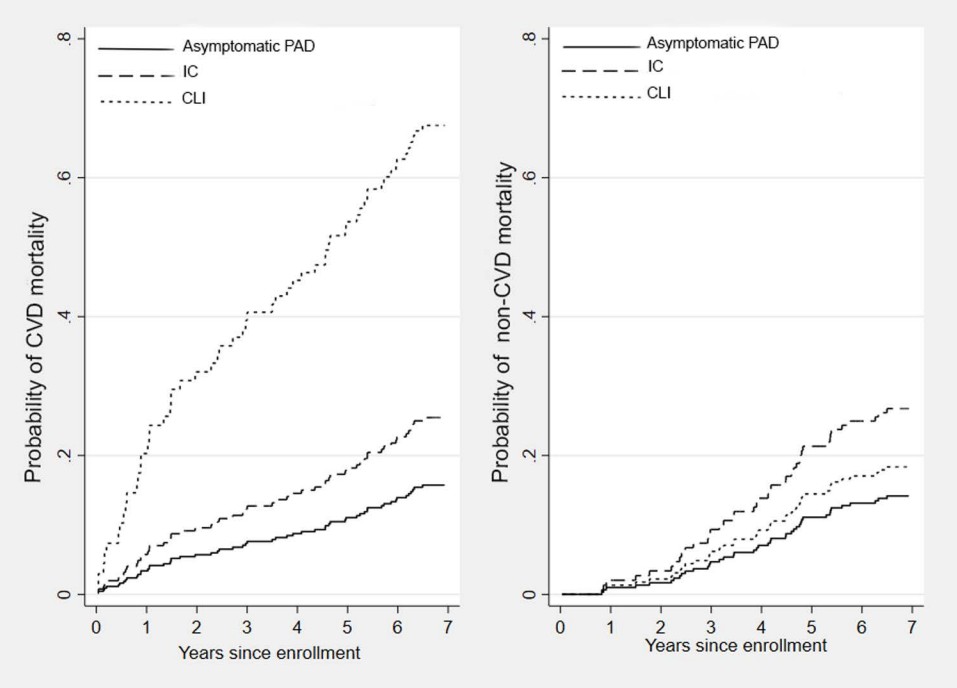

**Fig 3. Competing risk curves depict the cumulative probability of time to death for participants with T2DM and PAD stratified by PAD severity.** (A) The left panel shows the probabilities of CVD mortality. Patients with CLI had the highest event rates compared to the other groups (p < .05). However, the rate of CVD mortality did not significantly differ between those with IC and asymptomatic PAD (p = .37). (B) The right panel shows the competing risk event, i.e., other causes of death occurring before the CVD mortality. There was no significant difference in the non-CVD mortality rates among the three groups of participants.

increased mortality risk, appears to be consistent across various age groups and genders, primarily influencing long-term mortality outcomes [28,29].

The study further highlights the increased mortality risk associated with the underweight category in the PAD population, with prior research supporting this finding and positioning a lower BMI as a prognostic indicator of poor survival outcomes in PAD [28–30]. In contrast, the mortality risk for obese PAD patients did not significantly differ from that of their normal-weight or overweight counterparts, suggesting that the combination of T2DM and PAD may lead to complex systemic atherosclerotic and metabolic dysregulations [29]. Moreover, the association of advanced atherosclerosis with conditions such as ischemic heart disease, cerebrovascular disease, and severe PAD could contribute to malnutrition, muscle wasting, and bone loss, further complicating the ability to augment BMI [31,32]. The mechanisms underlying the increased mortality risk in underweight PAD patients, especially in the presence of T2DM, remain unclear, emphasizing the need for further research to guide targeted weight management interventions.

Additionally, this study identified a protective effect of insulin-sensitizing hypoglycemic drugs on overall mortality, but not specifically on CVD mortality, in PAD patients. Insulin sensitizers, including thiazolidinediones and biguanides, reduce insulin resistance and may slow atherosclerosis progression [33]. While randomized controlled trials have shown that these drugs could reduce all-cause mortality and non-fatal cardiovascular events in T2DM patients [34], some studies have reported an increased risk of heart failure [35]. Research specifically examining mortality outcomes in T2DM patients with PAD using insulin sensitizers remains limited. Our findings suggest potential benefits of insulin sensitizers beyond glycemic control in this high-risk population, though further research is needed to confirm these effects.

## Limitations

This investigation primarily focused on the impact of glycemic control therapy on mortality, while the potential effects of treatments targeting blood pressure and hyperlipidemia were not addressed. The exclusion of patients with recent cardiovascular events (within 3 months of enrollment), though necessary for establishing a stable baseline population, may have underestimated the full spectrum of cardiovascular risk in PAD patients.

Our study population was derived from tertiary medical centers, potentially overrepresenting older patients with advanced comorbidities and limiting generalizability to broader populations. Another significant limitation was the lack of systematic collection of activities of daily living (ADL) data at enrollment. Nursing records indicated that most participants (n = 268) were ambulatory, while some arrived at the clinic using a wheelchair (n = 4) or a stretcher (n = 6), primarily those with CLI. However, the lack of comprehensive functional status data limited our ability to analyze ADL as a potential predictor of mortality.

## Conclusion

This study's findings indicate that patients with T2DM and PAD, particularly those presenting with CLI, face a higher risk of both CVD-related and all-cause mortality. The data reveal that even asymptomatic PAD in diabetic individuals carries a mortality risk comparable to that observed in IC. This finding emphasizes the need for comprehensive treatment strategies that do not discriminate based on symptomatic presentation. These findings support the integration of PAD as a marker of systemic atherosclerosis within the diabetic care continuum, highlighting the importance of routine vascular screening and management in this high-risk population. Future investigations should focus on identifying the full spectrum of modifiable risk factors affecting mortality in patients with T2DM and PAD to optimize therapeutic approaches.

## Supporting information

**S1 Table. Risk factors management [a] at baseline of participants with type 2 diabetes and peripheral arterial disease (PAD). (N = 278).**
(DOCX)

## Acknowledgments

Language assistance during the manuscript editing process was provided by ChatGPT-4 (OpenAI, San Francisco, CA, USA) and Claude 2.1 (Anthropic, San Francisco, CA, USA), which helped refine the language, improve readability, and enhance overall coherence.

## Author contributions

**Conceptualization:** Amaraporn Rerkasem, Ampica Mangklabruks, Supawan Buranapin, Kiran Sony, Nimit Inpankaew, Rath Rerkasem, Kittipan Rerkasem.

**Formal analysis:** Amaraporn Rerkasem, Kittipan Rerkasem.

**Investigation:** Amaraporn Rerkasem, Ampica Mangklabruks, Supawan Buranapin, Kiran Sony, Nimit Inpankaew, Rath Rerkasem, Sasinat Pongtam.

**Methodology:** Amaraporn Rerkasem, Ampica Mangklabruks, Supawan Buranapin, Sasinat Pongtam, Kittipan Rerkasem.

**Supervision:** Kittipan Rerkasem.

**Writing – original draft:** Amaraporn Rerkasem, Kochaphan Phirom, Kittipan Rerkasem.

**Writing – review & editing:** Kochaphan Phirom.

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
