## [Decision Letter · Decision Letter 0]

12 Jan 2025

PONE-D-24-50975Incidence and predictors of cardiovascular disease mortality and all-cause mortality in patients with type II diabetes with peripheral arterial diseasePLOS ONE

Dear Dr. Rerkasem,

Thank you for submitting your manuscript to PLOS ONE. After careful consideration, we feel that it has merit but does not fully meet PLOS ONE’s publication criteria as it currently stands. Therefore, we invite you to submit a revised version of the manuscript that addresses the points raised during the review process.

We look forward to receiving your revised manuscript.

Kind regards,

Shukri AlSaif

Academic Editor

PLOS ONE

Journal Requirements:

Reviewers' comments:

Reviewer's Responses to Questions

**Comments to the Author**

1. Is the manuscript technically sound, and do the data support the conclusions?

Reviewer #1: Partly

2. Has the statistical analysis been performed appropriately and rigorously? 

Reviewer #1: I Don't Know

3. Have the authors made all data underlying the findings in their manuscript fully available?

Reviewer #1: Yes

4. Is the manuscript presented in an intelligible fashion and written in standard English?

Reviewer #1: Yes

5. Review Comments to the Author

Reviewer #1: PADs are classical but uprising diseases and the predictors are more to be focused.

major;

Why the recently diagnosed CI or CAD patients were excluded ?

While we are researching natural course of the PAD, I think this part is absolutely important. Also, Table1 contains history of OMI, old CI, and TIA. If you are focusing only "future (newly complicated)" CI or CAD, the title need some arrangement.

Do authors have any idea about medical treatment backgrounds ? There are only about insulins and DPP4 (Table).

minor;

1) ABI cannot stage severity of the PAD. (abstract part)

2) Check acronyms. DM or T2DM (better to unify), several acronyms are spelled out twice or more.

3) PADs are classified as asymptomatic, intermittent claudication, and CLI. Is the rest pain included as CLI ?

Personally, I want to know the difference between ulcer and rest pain.

4) Any idea for the ADL ? (walking/chair-bound/bed-bound etc.)

5) Any data for the hemodialysis patient ?

6) Any idea for the location of the PAD ? (iliac/femo-pop/BK)

6. PLOS authors have the option to publish the peer review history of their article (what does this mean? ). If published, this will include your full peer review and any attached files.

**Do you want your identity to be public for this peer review?** For information about this choice, including consent withdrawal, please see our Privacy Policy .

Reviewer #1: No

---

## [Author Response · Author response to Decision Letter 1]

27 Jan 2025

Dear Dr. Shukri AlSaif (Academic Editor)

Thank you for giving us the opportunity to revise and re-submit our manuscript to PLOS ONE.

We extend our sincere gratitude to you and the reviewers for the detailed and constructive feedback. In response, we have made significant revisions to the manuscript as suggested. Please find below our detailed responses to your and the reviewers' comments. All pages refer to the revised manuscript. In the tracked changes version, revisions in response to the reviewers and editors' comments are highlighted in yellow.

We hope this revised version meets the journal's standards and is now suitable for publication. However, we are open to making any further revisions that may be necessary. We look forward to hearing from you again in due course.

Kind regards,

Amaraporn Rerkasem, on behalf of all authors

REVIEWER #1

PADs are classical but uprising diseases and the predictors are more to be focused.

Reply: We appreciate the thoughtful comments from the reviewers regarding our manuscript. We have addressed each point as follows:

Major:

1) Why the recently diagnosed CI or CAD patients were excluded ?

Reply: Thank you for this important question. Regarding the exclusion of recently diagnosed cerebrovascular and coronary artery disease patients, this methodological decision was made to establish a stable baseline population where measurements reflect steady-state disease rather than acute post-event changes. The 3-month exclusion window helps minimize the confounding effects of immediate post-event treatments and temporary changes in cardiovascular risk factors that typically occur after acute events. This approach allows for a more reliable assessment of PAD's natural progression and its relationship with future cardiovascular outcomes. We have clarified this in our Study Population Subsection under Material and Methods Section and added the specific 3-month timeframe, which was not explicitly stated in our original manuscript now read as follows:

“The exclusion criteria were as follows: patients with a new diagnosis of cerebrovascular or coronary events within three months of enrollment to ensure baseline measurements reflected stable disease states rather than acute post-event changes. Individuals with stage III-IV malignancy or terminal cancer and a life expectancy less of than one year, as well as those with active HIV infection or AIDS-related vasculopathy were also excluded. Furthermore, patients requiring scheduled vascular surgery within the following six months —such as those with symptomatic abdominal aortic aneurysms or critical limb ischemia—were not included. Finally, those with systemic infections affecting the vasculature, such as fungal arteritis, were excluded from study.”

2) While we are researching natural course of the PAD, I think this part is absolutely important. Also, Table1 contains history of OMI, old CI, and TIA. If you are focusing only "future (newly complicated)" CI or CAD, the title need some arrangement.

Reply: We appreciate this astute observation. Concerning the presence of historical cardiovascular events in Table 1 while excluding recent events, this reflects our intent to capture the real-world comorbidity burden in the PAD population while avoiding the confounding effects of acute disease states. Our current title appropriately reflects the study's focus on mortality outcomes in this population, regardless of their cardiovascular history. We have clarified the distinction between included historical events and excluded recent events in our Methods Section (in reply of question #1) and acknowledged in our limitations as follows:

“The exclusion of patients with recent cardiovascular events (within 3 months of enrollment), though necessary for establishing a stable baseline population, may have underestimated the full spectrum of cardiovascular risk in PAD patients. ”

3) Do authors have any idea about medical treatment backgrounds? There are only about insulins and DPP4 (Table).

Reply: Thank you for highlighting the importance of medical treatment backgrounds. We agree that cardiovascular medications known to affect mortality in PAD patients should be considered in our analysis. Following your comment, we conducted additional analyses incorporating key cardiovascular medications based on their clinical relevance, observed distribution differences between study groups, and established effects on mortality from literature. These included antithrombotic agents, statin therapy, and RAAS blockers (combined ACEI/ARBs) alongside diabetes medications.

This expanded analysis revealed insulin sensitizer therapy as a protective factor for all-cause mortality, while other previously identified risk factors (critical limb ischemia, being underweight, and chronic kidney disease) remained significant predictors. The predictors of cardiovascular disease-related death remained consistent with our original findings.

We have revised Tables 4 and 5 to reflect these analyses and added a new discussion paragraph that provides a comprehensive understanding of how insulin sensitizer therapy influences mortality outcomes in our study population.

“Additionally, this study identified a protective effect of insulin-sensitizing hypoglycemic drugs on overall mortality, but not specifically on CVD mortality, in PAD patients. Insulin sensitizers, including thiazolidinediones and biguanides, reduce insulin resistance and may slow atherosclerosis progression [33]. While randomized controlled trials have shown that these drugs could reduce all-cause mortality and non-fatal cardiovascular events in T2DM patients [34], some studies have reported an increased risk of heart failure [35]. Research specifically examining mortality outcomes in T2DM patients with PAD using insulin sensitizers remains limited. Our findings suggest potential benefits of insulin sensitizers beyond glycemic control in this high-risk population, though further research is needed to confirm these effects.”

Minor:

1) ABI cannot stage severity of the PAD. (abstract part)

Reply: Thank you for your astute observation regarding ABI and PAD staging. You are correct that ABI alone cannot stage PAD severity. We have revised the abstract to clarify that PAD severity was determined through both clinical assessment and ABI measurements.

“These patients completed health questionnaires and underwent physical examinations including ankle-brachial index (ABI) measurements and clinical assessment to determine PAD severity.”

2) Check acronyms. DM or T2DM (better to unify), several acronyms are spelled out twice or more.

Reply: Thank you for your comment. We have carefully reviewed all acronyms in the manuscript and ensured consistency in their usage. Specifically, we have standardized the use of T2DM throughout the text instead of alternating between DM and T2DM. Additionally, we have removed redundant definitions of acronyms that were spelled out more than once. We appreciate your suggestion, which has helped improve the clarity and consistency of our manuscript.

3) PADs are classified as asymptomatic, intermittent claudication, and CLI. Is the rest pain included as CLI ? Personally, I want to know the difference between ulcer and rest pain.

Reply: Thank you for this important question about PAD classification. In our study, we classified rest pain as part of critical limb ischemia (CLI) following the TASC II 2007 guidelines. This classification reflects that both rest pain and tissue loss represent advanced stages requiring similar management approaches, distinct from intermittent claudication and asymptomatic PAD. Additionally, given the relatively small number of events in our real-world data, combining these presentations into the CLI category enabled more robust statistical analysis while maintaining clinical relevance. We have expanded the definition of CLI alongside the TASC II reference in the introduction section to provide better clarity.

“Critical limb ischemia (CLI), an advanced stage of PAD characterized by rest pain, ischemic ulcer, or gangrene [8], often results in amputation and death. [8]”

4) Any idea for the ADL ? (walking/chair-bound/bed-bound etc.)

Reply: Thank you for raising this important point about ADL status. While ADL was not systematically collected at enrollment as it was not part of our primary objectives, we did find in nursing records that 268 participants were ambulatory while 10 were chair-bond and bed-bound, with bed-bound status mainly occurring in CLI patients. We acknowledge this as a study limitation in the revised manuscript, as functional status could be an important mortality predictor in PAD patients. Future studies should systematically assess ADL status to better understand its relationship with mortality in this population. A limitation statement in the Limitations Section of the manuscript, now read as follows:

“Another significant limitation was the lack of systematic collection of activities of daily living (ADL) data at enrollment. Nursing records indicated that most participants (n=268) were ambulatory, while some arrived at the clinic using a wheelchair (n=4) or a stretcher (n=6), primarily those with CLI. However, the lack of comprehensive functional status data limited our ability to analyze ADL as a potential predictor of mortality.”

5) Any data for the hemodialysis patient ?

Reply: Thank you for inquiring about hemodialysis patients. Our study included 6 patients (2.2%) on maintenance hemodialysis at baseline. While hemodialysis is known to be associated with increased CVD mortality risk, our small sample size precluded robust statistical analysis of this relationship. So, we assessed kidney function primarily through eGFR<60 ml/min/1.73m² to capture the broader spectrum of CKD in our cohort. We have added hemodialysis details to the second paragraph of the Results section after the CKD findings (in Table 1), which now reads:

“Of the patients with CKD, 6 patients were on maintenance hemodialysis at baseline. Among these hemodialysis patients, 6 died during follow-up (3 from CVD-related causes, 3 from non-CVD causes).”

6) Any idea for the location of the PAD ? (iliac/femo-pop/BK)

Reply: Thank you for your comment. Identifying the specific anatomical location of PAD lesions requires advanced angiographic imaging (e.g., DSA, CTA, MRA), which, per the European guideline 2019, is typically reserved for patients requiring revascularization. Non-invasive methods like physical examination and ABI measurements are insufficient for precise localization.

In our study, 90% (251 of 278) of patients presented with asymptomatic PAD or claudication, effectively managed with conservative treatments, and thus did not require angiographic evaluation. Among the 10% (27 of 278) with critical limb ischemia (CLI), angiographic evaluation was performed in 5 cases where revascularization was feasible. These lesions were primarily in the femoropopliteal and tibial segments, with iliac involvement noted in four cases. The remaining CLI patients were unsuitable for revascularization due to extensive necrosis, gangrene, or severe comorbidities, and angiographic evaluations were not performed.

To address this further, we revised the inclusion criteria and added a new paragraph in the Study Population subsection of the Methods section to clarify the rationale for the limited lesion location data. The updates are as follows:

Inclusion criteria: "…Inclusion criteria required a diagnosis of T2DM, age ≥45, and PAD confirmed by symptoms or an ankle-brachial index (ABI) ≤0.9. Imaging techniques, such as duplex ultrasound, digital subtraction angiography (DSA), computed tomography angiography (CTA), or magnetic resonance angiography (MRA) were used for confirmation in selected cases when clinically indicated. All participants were required to have been under hyperglycemic control for at least six months before enrollment. "

New paragraph: "Due to the inclusion criteria and clinical characteristics of the study population, the anatomical location of PAD was not uniformly determined. Advanced imaging techniques (e.g., DSA, CTA, MRA) were reserved for cases where revascularization was clinically necessary, in line with European guidelines [11]. Initial assessments relied on physical examination, ABI measurements, and duplex ultrasonography, which are sufficient for diagnosing PAD but not for detailed anatomical localization of lesions."

---

## [Decision Letter · Decision Letter 1]

28 Feb 2025

PONE-D-24-50975R1Incidence and predictors of cardiovascular disease mortality and all-cause mortality in patients with type II diabetes with peripheral arterial diseasePLOS ONE

Dear Dr. Rerkasem,

Thank you for submitting your manuscript to PLOS ONE. After careful consideration, we feel that it has merit but does not fully meet PLOS ONE’s publication criteria as it currently stands. Therefore, we invite you to submit a revised version of the manuscript that addresses the points raised during the review process.

We look forward to receiving your revised manuscript.

Kind regards,

Shukri AlSaif

Academic Editor

PLOS ONE

Journal Requirements:

Reviewers' comments:

Reviewer's Responses to Questions

**Comments to the Author**

1. If the authors have adequately addressed your comments raised in a previous round of review and you feel that this manuscript is now acceptable for publication, you may indicate that here to bypass the “Comments to the Author” section, enter your conflict of interest statement in the “Confidential to Editor” section, and submit your "Accept" recommendation.

Reviewer #1: (No Response)

2. Is the manuscript technically sound, and do the data support the conclusions?

Reviewer #1: Partly

3. Has the statistical analysis been performed appropriately and rigorously? 

Reviewer #1: I Don't Know

4. Have the authors made all data underlying the findings in their manuscript fully available?

Reviewer #1: Yes

5. Is the manuscript presented in an intelligible fashion and written in standard English?

Reviewer #1: No

6. Review Comments to the Author

Reviewer #1: Thank you for the revision. Unfortunately, there are still several comments. The paper became much better than before.

It is minor revision but need to revise.

Acronyms, still need check. T2DM, CVD are spelled out more than twice in the main article. Does Myocardial infarction (MI) need acronym ? Besides, several acronyms such as CKD, COPD are need to spell out. Please check again and also need to check in Tables, Figure legends.

Table 1 is annoying. why those medications are not only medication name ? (use ? therapy ??) .

Also in another Table. Unify such as beta-blocker, antithrombotic, diuretics, statin...

dual antiplatelets or single is missing. Don't you have any information about details about it ?

ASA, P2Y12, also OACs ??

If you combine RAAs inhibitor on Table 4 and 5, why do you separate ACEI and ARB on Table 1.

what is "S1 Table" ?

In figure legend, intermittent claudication (CL) ?

7. PLOS authors have the option to publish the peer review history of their article (what does this mean? ). If published, this will include your full peer review and any attached files.

**Do you want your identity to be public for this peer review?** For information about this choice, including consent withdrawal, please see our Privacy Policy .

Reviewer #1: No

---

## [Author Response · Author response to Decision Letter 2]

16 Mar 2025

Shukri AlSaif (Academic Editor)

Thank you for submitting your manuscript to PLOS ONE. After careful consideration, we feel that it has merit but does not fully meet PLOS ONE’s publication criteria as it currently stands. Therefore, we invite you to submit a revised version of the manuscript that addresses the points raised during the review process.

Dear Dr. AlSaif

Thank you for giving us the opportunity to revise and re-submit our manuscript to PLOS ONE.

We extend our sincere gratitude to you and the reviewers for the detailed and constructive feedback. In response, we have made minor revisions to the manuscript as suggested. Please find below our detailed responses to your and the reviewers' comments. In the tracked changes version, revisions in response to the reviewers and editors' comments are highlighted in yellow.

We hope this revised version meets the journal's standards and is now suitable for publication. However, we are open to making any further revisions that may be necessary. We look forward to hearing from you again in due course.

Kind regards,

Kittipan Rerkasem, on behalf of all authors

REVIEWER #1

Thank you for the revision. Unfortunately, there are still several comments. The paper became much better than before. It is minor revision but need to revise.

Reply: Thank you for your positive feedback and for recognizing the improvements in our revision. We appreciate your detailed review and have carefully addressed the remaining comments to further enhance the clarity and quality of the manuscript.

We have revised the manuscript accordingly, ensuring that all suggested changes have been incorporated. Please find our point-by-point responses below, along with the corresponding revisions in the manuscript.

We sincerely appreciate your time and effort in reviewing our work and believe that these refinements strengthen the overall quality of the paper.

Minor:

1) Acronyms, still need check. T2DM, CVD are spelled out more than twice in the main article. Does Myocardial infarction (MI) need acronym ? Besides, several acronyms such as CKD, COPD are need to spell out. Please check again and also need to check in Tables, Figure legends.

Reply: Thank you for your careful review and for pointing out the inconsistencies in acronym usage. We have now thoroughly checked the manuscript, including the main text, tables, and figure legends, to ensure that acronyms are introduced only once and spelled out where necessary.

• T2DM (Type 2 Diabetes Mellitus) and CVD (Cardiovascular Disease) have been reviewed to ensure they are not spelled out multiple times inappropriately.

• Myocardial Infarction (MI) has been checked, and we have ensured that the acronym is introduced properly and used consistently.

• Chronic Kidney Disease (CKD) and Chronic Obstructive Pulmonary Disease (COPD) are now spelled out at first mention in both the text and tables/figures.

We appreciate your valuable feedback and believe these revisions improve clarity and consistency in the manuscript.

2) Table 1 is annoying. why those medications are not only medication name ? (use ? therapy ??) .

Also in another Table. Unify such as beta-blocker, antithrombotic, diuretics, statin...

dual antiplatelets or single is missing. Don't you have any information about details about it ?

ASA, P2Y12, also OACs ??

If you combine RAAs inhibitor on Table 4 and 5, why do you separate ACEI and ARB on Table 1.

Reply: Thank you for your insightful comments regarding Table 1 and the consistency of medication classification across tables. We have carefully revised the tables to improve clarity and standardization based on your suggestions.

• Medication Naming: We have revised the presentation of medications to list only the medication names under the category of medication use for clarity and consistency.

• Antithrombotic: we have now included a more detailed analysis, specifying dual and single antiplatelet therapy, as well as oral anticoagulants (OACs), to provide a more comprehensive overview.

• RAAS Inhibitor Consistency: To maintain uniformity, we have aligned Table 1 with Tables 4 and 5, ensuring that RAAS inhibitors are present consistently.

We appreciate your detailed feedback, which has helped refine the presentation of our data.

3) what is "S1 Table" ?

Reply: We apologize for the oversight. "S1 Table" refers to the supplementary table, which was mistakenly omitted in the previous version. We have now added it at the end of the manuscript in the Supporting Data section. Thank you for pointing this out.

4) In figure legend, intermittent claudication (CL) ?

Reply: Thank you for pointing this out. We apologize for the misspelling of intermittent claudication (IC) as intermittent claudication (CL). We have corrected this in both the figure and the figure legend to ensure accuracy and consistency.

---

## [Editor Report · Decision Letter 2]

23 Mar 2025

Incidence and predictors of cardiovascular disease mortality and all-cause mortality in patients with type II diabetes with peripheral arterial disease

PONE-D-24-50975R2

Dear Dr. Rerkasem,

We’re pleased to inform you that your manuscript has been judged scientifically suitable for publication and will be formally accepted for publication once it meets all outstanding technical requirements.

Kind regards,

Shukri AlSaif

Academic Editor

PLOS ONE
---

## [Editor Report · Acceptance letter]

PONE-D-24-50975R2

PLOS ONE

Dear Dr. Rerkasem,

I'm pleased to inform you that your manuscript has been deemed suitable for publication in PLOS ONE. Congratulations! Your manuscript is now being handed over to our production team.

Kind regards,

on behalf of

Dr. Shukri AlSaif

Academic Editor

PLOS ONE